# Coordination between Surface Lattice Resonances of Poly(glycidyl Methacrylate) Line Array and Surface Plasmon Resonances of CdS Quantum on Silicon Surface

**DOI:** 10.3390/polym11030558

**Published:** 2019-03-25

**Authors:** Shuenn-Kung Su, Feng-Ping Lin, Chih-Feng Huang, Chien-Hsing Lu, Jem-Kun Chen

**Affiliations:** 1Department of Materials Science and Engineering, National Taiwan University of Science and Technology, Taipei 106, Taiwan; sksu@mail.ntust.edu.tw (S.-K.S.); D10304007@mail.ntust.edu.tw (F.-P.L.); 2Department of Chemical Engineering, National Chung Hsing University, Taichung 40227, Taiwan; 3Department of Obstetrics and Gynecology, Taichung Veterans General Hospital, Taichung 40705, Taiwan; 4Ph. D. Program in Translational Medicine, and Rong Hsing Research Center for Translational Medicine, National Chung Hsing University, Taichung 40227, Taiwan; 5Taiwan Building Technology Center, National Taiwan University of Science and Technology, Taipei 10607, Taiwan

**Keywords:** PGMA brush, CdS quantum dots, one-dimensional grating

## Abstract

In this work, a unique hybrid system is proposed for one-dimensional gratings comprising of poly(glycidyl methacrylate) (PGMA) brushes and CdS quantum dots (CQDs). Generally, the emission of QDs is too weak to be observed in a dry state. Plasmonic resonances of the grating structures can be used to enhance the light emission or absorption of CQDs. The interaction between PGMA plasmonic nanostructures and inorganic CQDs plays a crucial role in engineering the light harvest, notably for optoelectronic applications. Extinction measurements of the hybrid system consisting of a PGMA grating and CQDs are reported. We designed one-dimensional gratings with various resolutions to tune the absorptance peaks of grating. PGMA grating grafted from a 1.5 µm resolution of trench arrays of photoresist exhibited absorptance peak at 395 nm, close to the absorption peak of CQDs, resulting in the photoluminescence enhancement of CQDs on the grating due to high charge carriers’ recombination rate. Generally, the emission of quantum dots occurs under irradiation at characteristic wavelengths. Immobilizing QDs on the grating facilitates the emission of QDs under irradiation of full-wavelength light. Furthermore, the PGMA gratings with CQDs were immersed in various solvents to change the geometries resulting the shift of absorptance peak of grating. The proposed method could be applied for sensing the nature of the surrounding media and vice versa, as well as for various media of solvents.

## 1. Introduction

Polymer brushes are defined as the surfaces to which polymer chains are attached to a substrate, signifying a class of polymeric coatings according to one of their margins through the covalent bond [1,2,3,4]. The fabrication of well-defined chemically patterned surfaces at the micro- and nanoscale is at the heart of many modern scientific and technical fields. To date, there are several well-patterned polymer brushes with specific control levels over the polymer brush composition and thickness, and patterns have been fabricated by semiconductor process methods such as photolithography [5,6], plasma etching [7,8], reversible addition-fragmentation chain transfer polymerization (RAFT) [9,10] and atom transfer radical polymerization (ATRP) [11,12,13,14]. The use of polymers as building blocks for surface modification gives rise to the preparation of “smart” or responsive surfaces based on conformational changes in the polymer backbones. Stable polymer brushes can provide excellent mechanical and chemical protection to substrates, alter the electrochemical characteristics of the interface, and provide new pathways for functionalization of silicon surfaces. [8,15]. One particular advantage that polymer brushes have over spin-coated polymer layers is their stability against solvents or harsh conditions (e.g., high temperatures) because they are bonded covalently to the substrates. One attractive strategy involves the application of a functional group polymer coatings that can subsequently be used as a platform to introduce other active molecules. Poly(glycidyl methacrylate) (PGMA) is the most famous functional polymer with the unique characteristic feature easy ring opening reactions with −NH_2_, −OH, −COOH and −SH, resulting in a different functional PGMA derivative. In addition, PGMA functional polymer has plenty of pendant epoxy groups. The polymerization process of ATRP and RAFT [16,17,18] offers excellent structural tunability, with the additional advantage of low-cost monomer source for constructing the PGMA brushes, which is a new kind of brush for materials science or useful surface linker and biochemistry [19,20,21,22].

Plasmonic nanostructures acting as optical antennas have been thoroughly investigated for different purposes in recent years, because of their abilities to focus light on subwavelength volumes and to create strongly enhanced near fields on their surfaces. Such nanostructures show the high promise for applications in the areas of near-field optical microscopy and spectroscopy, biosensing, or light harvesting in optoelectronic devices [23]. Optical antennas can enhance the spontaneous emission or the light absorption of organic molecules in their vicinity, which have been related to their localized surface plasmon resonances (LSPRs) [24,25]. In chains or arrays of plasmonic nanostructures, collective grating-induced modes, sometimes called SLRs (surface lattice resonances), can also be of use, as can the LSPRs of single particles [26,27]. Plasmonic gratings have been combined with organic dyes and semiconductors or dipolar emitters for experimental and theoretical study of the spontaneous emission in coupled hybrid systems [28]. The emission of dye molecules can be strongly enhanced and directed in such systems when they radiatively decay through the collective grating modes [29]. Furthermore, stimulated emission in such hybrid systems has also been observed [30]. In addition to the emission enhancement, the coupling of lattice resonances to molecules has also been studied theoretically and experimentally [31,32]. The one/two-dimensional grating structures characteristic feature of selective wavelength absorption properties has attracted substantial attention in metal ion and DNA sensors applications. [33,34], which demand defined absorptance in the specified spectral region and low absorptance in others. There are many reports on the investigations of the absorptance properties in various surface geometries of micro/nanostructures. Plasmonic gratings, combined with quantum dots for experimental study on coupled hybrid systems, to investigate their optical properties, have not been widely reported.

In this work, we fabricated one-dimensional plasmonic PGMA gratings (1DPPGs) in order to undertake an impact study on the grating-induced effects on the optical properties of a hybrid system containing both surface lattice resonances (SLRs) of PGMA gratings and surface plasmon resonances (SPRs) of CdS quantum dots (CQDs). 1DPPGs offer the advantage of tailoring the plasmonic and lattice modes through geometrical parameters and spectral tuning by the width and the period of the lines. The employed procedure makes it possible to graft PGMA chains onto the emphasized substrate in a brush formation as an absorber to immobilize CQDs on the 1DPPGs. The selective-wavelength absorptance of 1DPPG provides extra excitation to the surface plasmon polaritons (SPPs) in order to achieve photoluminescence (PL) enhancement of CQDs. The 1DPPGs grafted from 1 µm, 1.5 µm, 2 µm and 3 µm resolution trench arrays of photoresist at 1.5 duty ratio were chosen because the absorptance peaks ranged from 371 to 459 nm, making them close to those of CQDs. The results showed that the combination of 1DPPG and CQD substantially enhance optical extinction. In addition, the geometry of 1DPPG can be varied with solvents to optimize the intensity of the CQD PL peak. The coordination between SLRs and SPRs offers greater opportunities when considering tunable optical outcomes in the nanosensor applications.

## 2. Experimental Section

### 2.1. Materials

The single-crystal grown silicon wafers were procured with thickness and diameter of 1.5 nm and 6 inches respectively from Hitachi, Inc. (Tokyo, Japan) with the preferred orientation of 100. They were then cut into substrates of 1 cm × 1 cm for the experiments. Notable impurities, comprising organic contaminants and dust particles, were effectively removed by following the experimental procedures described in our previous publications [35]. Acros Organics (Geel, Belgium) supplied 2-Bromo-2-methylpropionyl bromide (BB), 3-Aminopropyltriethoxysilane (AS), triethylamine (TA), glycidyl methacrylate (GMA), copper(I) bromide (Cu(I)Br), copper(II) bromide (CuBr_2_), and 1,1,4,7,7-pentamethyldiethylenetriamin (PMDETA) for the process of graft polymerization. It is important to note that GMA, PMDETA, AS and BB underwent a vacuum distillation process prior to use. Acros Organics also supplied cadmium chloride hemi (pentahydrate), sodium sulfide nonahydrate and cysteamine for the synthesis of CQD. All other chemicals and solvents used in the experiments were supplied by the Aldrich Chemicals (Darmstadt, Germany), and were of reagent grade quality; these chemicals were used without further processing.

### 2.2. Line Array Patterned Polymer Brushes

The approach used for the fabrication of the grating employs a very-large-scale integration (VLSI) process is already discussed in our previous report [36]. The procedure of the fabrication method is schematically shown in Figure 1. Step A: The photoresist, of a thickness of nearly 780 nm, was spun onto the surface of HMDS-treated Si wafer. Step B: For the patterned photoresistant, I-line lithography was utilized for making trench arrays with resolutions of 1 µm, 1.5 µm, 2 µm and 3 µm after the duty ratio of 1.5. Step C: On the surface of the Si substrate, the selective units of AS were carefully assembled onto the bare regions of the substrate, which were marked as Si-AS. Step D: After step C, the immersion of the sample in the solution containing 2% (*v*/*v*) solution of TA in tetrahydrofuran (THF) and BB at the temperature of 20 °C for 8 h yielded a surface with regions of halogen groups for ATRP, which was denoted as Si-AS-BB. At the same time, rinsing with a solvent helped to remove the residual photoresist from the HMDS-treated surface which were left behind. Step E: As the next step, ATRP was utilized for grafting the patterned PGMA brushes onto the initiator-modified surface of Si. For the preparation of the PGMA brushes on the surface of Si-AS-BB, a mixture of methanol and water was prepared in a ratio of 2:1, and chemicals PMDETA, Cu(I)Br, CuBr_2_ and GMA were added into it. The Si-AS-BB substrate was dropped into the deoxygenated solution under an Ar atmosphere at a temperature of 90 °C for 15 min. After 12 h of polymerization, the unreacted monomer, non-grafted material, and catalyst were detached by placing the wafers into a Soxhlet apparatus. The samples were dried under vacuum at a temperature of 80 °C for 20 min, and was denoted as Si-PGMA. Step F: The preparation and characterization of the cysteamine-capped CQDs used in this work have been described in an earlier report [37]. The samples of 1DPPG were placed in a round-bottomed flask containing DMF, and were heated under stirring at a temperature of 80 °C. Then, the excess cysteamine-modified CQDs were transferred into the system to initiate the reaction, which continued for 12 h, to obtain the CQD-modified 1DPPG. The samples were repeatedly rinsed with DMF to remove the non-adsorbed or weakly-adsorbed particles. The products were removed from the solution for cleaning with diethyl ether. The end product, obtained by vacuum drying in an oven for 24 h for further utilization, was denoted as Si-PGMA-CQD. In addition, these 1DPPGs, grafted from 1, 1.5, 2 and 3 µm resolution trench arrays of photoresists, were denoted as PG1, PG1.5, PG2 and PG3, respectively; after CQD immobilization, they were denoted as PQ1, PQ1.5, PQ2, and PQ3, respectively. PGMA was grafted onto a plane substrate without patterns for 24 h before and after CQD immobilization as controls, denoted as PGP and PQP, respectively, to observe the enhancement of CQDs by PGMA grafting. In addition, calibration curves of various fluorescence intensities at various CQD concentrations in the solution were obtained to evaluate the amount of CQD immobilization on the gratings of 1 cm × 1 cm in size. The surface concentrations of CQDs per cm square were calculated by the amount of change in the CQDs before and after the immobilization on the gratings. The saturated surface concentration of the CQDs on the PG1, PG1.5, PG2, PG3 and planer sample were ca. 84.6, 88.5, 96.3, 102.5 and 216.9 µg/cm^2^, respectively. In order to the optimize the optical properties of the PGMA-CQD gratings, the sample was immersed in various solvents including water, EtOH, toluene, THF, dioxane, toluene and CHCl_3_ for various times in order to tune the grating height. After this step, the samples were placed into a vacuum freeze dryer (BENCHTOP 2K; VIRTIS; New York, NY, USA) set at −40 °C for 10 min for solvent removal. Polymers can be dissolved using good solvents that are able to stretch the polymer chains; on the other hand, polymers cannot be dissolved in poor solvents. This results in polymer collapsing chains [38]. In this study, poor and good solvents for PGMA were water and dioxane, respectively. X-ray photoelectron spectroscopy (XPS) studies were carried out in Scientific Theta Probe, London, UK. The morphologies of lyophilized samples were characterized using an atomic force microscope (AFM; Veeco Dimension 5000 scanning probe microscope, Bruker, Billerica, MA, USA). A transmission electron microscope, supplied by Philips Tecnai G2 F20, was used for capturing transmission electron microscopy (TEM, Philips, Amsterdam, Nederland) images, operating at 200 kV an (accelerating voltage). Perkin Elmer Lambda 25 spectrophotometer (Bruker, Billerica, MA, USA) was used for measuring the UV-vis spectrum of the samples. The 3D distribution of the luminescent CQDs was captured using confocal laser scanning microscopy (CLSM, Leica TCS SP5, Leica, Wetzlar, Germany). The Ar laser with a wavelength of 395 nm and set to 10 mW was used as the excitation light to observe the luminescence of the samples.

### 2.3. Modeling Development and Hybrid Method

Optical antennas can enhance the spontaneous emission or the light absorption in their vicinity, a phenomenon which has been related to LSPRs [39,40]. In chains or arrays of plasmonic nanostructures, collective grating-induced modes, also known as SLRs, can occur in addition to the LSPRs of single particles [24,25]. These modes are highly related to Rayleigh anomalies, which will occur when the light is diffracted by a grating angle of 90°, i.e., parallel to the surface. The condition for the appearance of these anomalies is connected to the wave vector (k-vector) component of the light diffracted by the grating that is oriented perpendicular to the grating plane. The grating is defined in the x-y plane; therefore, this vector is parallel to the z-direction. The k-vector of the diffracted light can be written as [41]
(1)$$\vec{k}=\{k_{x},k_{y},k_{z}\}=\{k_{x0}+m_{x}\frac{2\pi }{p_{x}},k_{y0}+m_{y}\frac{2\pi }{p_{y}},\sqrt{n_{i}{}^{2}k_{0}{}^{2}-{\left(k_{x0}+m_{x}\frac{2\pi }{p_{x}}\right)}^{2}-{\left(k_{y0}+m_{y}\frac{2\pi }{p_{y}}\right)}^{2}}\}$$
with the initial incoming k-vector $\vec{k_{0}}$ = {*k_x_*_0_, *k_y_*_0_, *k_z_*_0_}, magnitude *k*_0_, the grating orders *m_x_*, *m_y_* = 0, ±1, ±2, …, and the periods of the grating in x and y direction are *P_x_*, *P_y_*. With this notation, grating anomalies can occur if *k_z_* = 0, i.e., if there is no propagation of the diffracted light normal to the plane of the grating. The asymmetric configurations with the different refractive indices *n_i_* above and below the grating are fulfilled twice per grating order, i.e., one on the upper surface side (*n_i_* = *n*_1_) and the other on the lower substrate side (*n_i_* = *n*_2_).

The schematic grating structure illustration considered in this study is shown in Figure 2. In the structure (Figure 2), the one-dimensional PGMA grating is assembled on the top opaque Si substrate. It should be remembered Si was chosen as the grating/substrate material because of its lightweight characteristic, inherent stability and widespread use for device applications. The subwavelength grating and substrate selection as the polymer material and binary profile grating structure have advantages over many other gratings presented in the previous literature in terms of cost and simplicity of the fabrication process [42,43]. The geometry of the grating structure presented in Figure 2 was characterized by three fundamental parameters, namely, period (*P*), the grating ridge width (*w*) and the grating thickness (*h*). The numerical model for absorptance determination is also shown. The energy balance for an opaque object implies that the value of absorptance can be calculated by the subtraction of reflectance value from unity.

In our current study, it is assumed that the incident light is travelling in the free space as a plane wave with an orientation determined by the angle among the surface normal (*z*) and wavevector (*k*). In addition, the plane which contains both the surface normal and incident light is defined as the plane of incidence. In the interests of simplicity, the only the *x*-*z* plane is considered for the analysis. The Lorentz-Drude model is used to explain the wavelength-dependent dielectric function ε of the grating and optical response analysis, as shown in Figure 2 [44,45]:(2)$$\varepsilon (\omega )=\varepsilon_{r,\infty }+\sum_{k=0}^{K}\frac{f_{k}\omega_{p}{}^{2}}{\omega_{k}{}^{2}-\omega^{2}+j\omega \Gamma_{k}}$$
where ε*_r_*_,∞_ = the optical dielectric constant at an infinite frequency and *ω* = 2π*c*_0_/λ = the speed, angular frequency and wavelength of the light in a vacuum, respectively. Finally, *ω_k_*_,_
*ω_p_*, *f_k_* and Γ*_k_* are the resonance frequency, plasma frequency, strength and damping frequency of the kth oscillator, respectively. In this study, experimental and theoretical principles are specified in the search range for each parameter, and the maximum subwavelength grating period is defined in the literature [46]. Similarly, the geometry constraints determine the value of filling factor in the order of 0 < *f* < 1 and the thickness-to-period ratio, 0.1 < *h*/*P* < 1, used for the grating thickness setting for both shallow and deep gratings.

## 3. Results and Discussion

### 3.1. Surface Properties of the PGMA-CQD Brushes

The homogeneously-distributed 1DPPGs were assembled with the silicon substrate using the familiar “grafting from” method [11]. The chemical compositions during each surface modification route were determined using XPS, as shown in Appendix A. XPS survey spectra of the various stages show the binding energies values of Si(2p), O(1s), and C(1s) as 100.0 eV, 530.0 eV, and 284.5 eV, in agreement with our previous literature [47]. Furthermore, Appendix A exhibits the binding energies (BE) with two peak components at around 99 eV and 103 eV; these may be attributed to Si-Si and Si-O species corresponding with the Si 2p core-level spectra of the Si(100) surface. A single peak component with a binding energy (BE) of approximately 402 eV appeared in the N 1s core-level spectrum of the Si-AS surface; this was attributed to N–H species. The Si-AS surface treated with BB was attached to the halogen group for ATRP and produced the Si-AS-BB surface. The Si surface was preferably halogen-ended after the treatment with BB and confirmed by the disappearance of the signal for the N–H species at 402 eV. The Si-AS-BB surface, terminated with the halogen on the initiator-functionalized Si surface, was confirmed by the weak signal of the N–C species with BE at 400 eV. Consequently, there was no signal appearance for the Si-AS surface prior to treatment with a BB core-level spectrum of the Br 3d. The N–H group is changed into a N–C group due to the treatment of the Si–AS with BB, and the Si surfaces obtained along with the Br species were confirmed by the BE signal which emerged from the core-level spectrum of the Si-AS-BB surface at 70 eV in the Br 3d. The elemental components of C and Br were examined after grafting the PGMA brushes onto the Si-AS-BB surface. The two peak components in the BE of approximately 405 eV and 164 eV, attributed to Cd^2+^ and S^2−^ species, appeared after the immobilization of CQDs. Appendix A displays the high-resolution C 1s region from Si-PGMA.

The fitted peaks centred at 284.8 eV, 286.8 eV and 288.7 eV correspond to C–C/H, C=O and O–C=O, respectively. The three peak component ratio areas were 3.19:2.98:1, matching the anticipated ratio of 3:3:1 for pure PGMA. The C, O ratio values of 7.1:2.8 for the Si-PGMA surface obtained from XPS analysis were in excellent agreement with the theoretical ratio (7:3). In Appendix A, the XPS spectrum of Cd^2+^ for the Si-PGMA-CQD nanocomposites showed doublet binding energies at ~405.3 eV and ~412.3 eV. These values largely matched reported values for Cd^2+^ in the literature [48] i.e., 405.1 eV & 412.1 eV for Cd 3d_5/2_ and Cd 3d_3/2_, respectively. The XPS spectra of S 2p peaks are shown in Appendix A, with a binding energy of 164.3 eV for the elemental sulfur [49]. These results confirm the formation of PGMA-CQD polymer brushes.

Figure 3 shows the 2D, 3D and a line cross-section atomic force microscopy (AFM) topographic image analysis of the gratings at 1.5 duty ratio. The PGMA-CQD brushes were grafted from the bottom surface of trench arrays photoresist using ATRPs as line arrays. As shown in Figure 3a, PQ1 was fabricated successfully on the Si surface; this occurred as a dense characteristic overlayer, with a line scale of nearly 1 µm inside a scanning area of 20 µm × 20 µm. The lines of PQ1 reveal irregular, hill-like structures with bottom and top widths of 2.1 µm and 286 nm respectively. The hill-like structures can be attributed to the collapse of the polymer brush because of the narrow grafting sites. Moreover, the line patterns of PQ1 have a uniform height (322 nm) and space (304 nm) among the lines. Note that the space among the lines represents the polished silicon surface that predominately reflects the incident light to generate diffraction. In addition, the grating period (P) is ca. 2.4 µm, which is the sum of width and space. The irregular hill-like structure is not obviously visible upon increasing the resolutions of the grafting sites from 1.5 µm to 3 µm (Figure 3b,c). Grating periods of PQ1.5, PQ2 and PQ3 were designed with spaces and widths of 2.886, 4.397, and 7.043 respectively in order to tune the absorption peak to a similar thickness (320 nm). The collapse of the polymer brushes and the inaccuracy of lithography, the scale of these spaces, widths and grating periods, did not match perfectly with the trench scale of the photoresist. The 1DPPGs can be considered as a medium mixed with PGMA and air, which is consistent with the effective medium theory. When the PGMA-functionalized line arrays absorbed the CQDs, we expected to observe a change in the values of the effective refractive index (*n*_eff_) on the gratings with the high refractive index to that of the CQDs. Using ellipsometry, the refractive index was measured on the PGMA thin film standard value at 633 nm to be 1.353. The grafting of PGMA as the line arrays reduced the value of *n*_eff_ from 1.353 to 1.193. The CQDs with a high refractive index (2.476) could absorb with the PGMA, resulting in an increase of *n*_eff_ for the CQDs absorbed-PGMA thin film by nearly 1.884. However, such a sudden large change, i.e., from 1.353 to 1.884, was not large enough. The liner structure of the PGMA decreased the refractive index, thus increasing the refractive index difference before and after the absorption of the CQDs.

### 3.2. Optical Properties of the Grating

Figure 4a shows the UV-vis absorptance spectra of bare CQDs. As can be seen, a single CdS presents a strong absorption band edge at 395 nm, which is related to the intrinsic band gap absorption. The band gap energy of the CQD was calculated to be nearly 3.13 eV. The inserts represent the TEM images and photography of CQDs, indicated particle sizes of ca. 4 nm and showed an orange color under UV irradiation. All the incident radiation was absorbed by a near unity absorber device at the operating frequency, while all the other propagation channels of light were disabled. The grating absorber, which is a special resonant electromagnetic wave absorber, used a reflective plane with an etched shallow periodic grid [50]. The periodic grid and incident radiation interactions created resonance, which in turn produced a period of anomalous diffraction [8]. The effect changes in the grating period and grating ridge width on the absorptance spectra of the PGMA grating, as shown Figure 4b, indicate that with a constant grating height, there is an increment in the grating ridge in the order of 2.1 µm to 4.4 µm, without any virtual changes in the value of peak absorptance and FWHM. The as-prepared PG1 exhibited an absorption peak at a wavelength of ca. 371 nm, which could be considered a resonant absorber. Furthermore, PG1 also partially absorbed a 280 nm wavelength, but the degree of absorptance at 280 nm was lower than that at 371 nm for PG1. Therefore, we select the absorptance at 370 nm as the marker to identify the shift of absorptance with the scale for the stripe pattern. In the case of PG1.5_,_ the absorption peak shifted to 395 nm, which is closer to the absorption peak of CQDs. Similarly, the absorption peaks of PG2 and PG3 were ca. 429 nm and 451 nm, respectively. Note that the PGMA-grafted sample without the strip patterns, denoted as PFP, did not show any absorptance. The light will propagate in all directions due to the action of the grating slits as quasi-point sources, while the resonant absorber is travelling with a normal incidence plane wave. After that, the light observed by the resonant absorber usually consists of the sum of interfering wave components, which could originate from each slit grating. The diffracted light will pass through the space in any of the slits, but at the same time, the path length in the grating for each slit will vary. The variation in the length of the path results in the formation of additive and destructive interference due to the addition or subtraction of wave phases to form the peaks and valleys. The inset of Figure 4b presents photographic images of the PG1, PG1.5, PG2, PG3 and PGP, recorded using a standard camera. The as-prepared PG1 exhibited a blue color, which could be regarded as a resonant absorber. Upon increasing the scale, the films gradually turned light green. The PFP film appeared slightly off-white without any color visualizing, indicating that the flat film did not possess any effect of resonant absorbance with lower absorption intensity.

The physical mechanism of enhanced transmission/absorption has been widely studied using micro/nanostructures for applications [42,51,52]. It is universally accepted that the Rayleigh-Wood anomaly (RW) is responsible for the physical mechanism happening in these grating structures or SPPs [53]. The grating angles at a certain disappearance or emergence of the diffraction orders in the RW anomaly induces a greater difference in the value of transmittance, reflectance and absorptance spectra by the subwavelength grating structures. The occurrence of RW can be predicted by the following equation:(3)$${\left(\frac{\lambda }{P}j\right)}^{2}+2\frac{\lambda }{P}\sin\theta -\cos^{2}\theta =0$$
where *λ* = incident wavelength; *P* = grating period; *θ* = angle of incidence and *j* = diffraction order. SPPs propagate as electromagnetic waves between the interfaces of dielectric and metals, due to the charge couplings oscillations of free electrons. The SPP excitation condition can be expressed as follows [54]:(4)$$n_{d}\sin\theta +j\frac{\lambda }{P}=\pm \sqrt{\frac{\varepsilon_{m}n_{d}{}^{2}}{\varepsilon_{m}+n_{d}{}^{2}}}$$
where *θ* = angle of incidence and *ε_m_* and *n_d_* are the permittivity (i.e., PGMA-CQD in the extant case) and the refractive index of air (*n_d_* = 1) respectively. Moreover, *λ* = wavelength (200~800 nm in the present case) and *j* = an integer indicating the diffraction order. Lastly, the “±“ sign corresponds to the diffraction orders of *j* > 0 and *j* < 0.

Photoluminescence (PL) emission spectra characterization was employed to investigate the fluorescence enhancement of CQD on the gratings. Generally, a PL with less intensity denotes the promoted charge separation, while a higher intensity implies a rapid charge recombination [55]. Figure 5a presents the contrast of the PL emission spectra of the PGMA-CQD gratings. The PL spectra of CQDs were characterized by a sharp peak around 580 nm, which, consistent with previous studies [56], can be attributed to a band-to-band emission of CQDs immobilized on a planer PGMA layer. An increase in fluorescence was observed for the CQDs immobilized on PGMA ODGs, suggesting an enhanced charge carrier recombination rate. The results of PL demonstrate that the combining of 1DPPG with CQD as a composite is favorable for the separation of photogenerated electrons and holes under light irradiation. The primary reasons for this are that the amounts of CQDs on the 1DPPG are different, and the intensity per CQD weight is defined to evaluate the enhancement of CQDs by the gratings (Figure 5b). We fabricated these gratings with a scale range from 1 to 3 um, according the previous reports, to match the absorptance peaks of the quantum dots. In our case, the PQ1.5 exhibits the highest fluorescence among the PGMA-CQD gratings, which could be attributed to the fact that the absorption peak of PQ1.5 matched perfectly with that of CQDs, thereby facilitating the recombination rate to the carrier. Figure 6 displays the CLSM images of the PGMA-CQD gratings. The apparent line arrays of red luminescence with various resolutions represent a good level of immobilization of CQDs on 1DPPG. The immobilization of CQDs on the polymer brushes was observed by manipulating the covalent bonding among epoxy functionalities of PGMA and surface (–NH groups) functionalities of CQDs. The placement of CQDs into the incubated aqueous solution resulted in homogeneous distribution with good dispersions on the PGMA brushes (Figure 6a–d). Consequently, the PQ1.5 exhibited the highest red luminescence among the PGMA-CQD gratings as well, which is consistent with the results of the PL emission spectra. A plasmonic absorber of grating on the silicon substrate could absorb the specific wavelengths at the interface to excite the CQDs intensively, resulting in a concentration effect of the fluorescence from CQD without scattering. On the other hand, light is mostly reflected from a flat silicon substrate without high absorption, resulting in a relatively low level of excitation of the CQDs. Figure 6e shows the image of CLSM images of PGMA-CQD thin film. Usually, the scattering effect facilitates the dispersion of the PGMA-CQD thin film fluorescence with partial non-uniformity.

In addition, in order to optimize the fluorescence of the grating, the PQ1.5 in the air is used as a standard sample to record the intensity change of the PL peak at 395 nm after immersion into various solvents. Generally, a polymer brush is swollen using a good solvent, resulting in a stretched regime. In contrast, the stretched regime of the tethered polymer is repressed in poor solvent treatment of the coiled regime in the line array layer. In this case, CHCl_3_, THF, dioxane and toluene are good solvents, while water and ethanol are poor for PGMA. In the method described above, solvent treatments of PQ1.5 lead to a reversible conformational change. The effective weakening and enhancing of the grating are motivated by the degree changes of the coiled regime of tethered PGMA. Furthermore, the immersion times of the change in grating height and absorptance peak in various solvents are shown in Figure 7. We observed approximately linear increases in the thickness of the PGMA grating upon increasing the immersion time to 30 s in various solvents (Figure 7a). The grating thickness reached a plateau after 40 s immersion time. The formation of the stretching regime of the PGMA brushes after good solvent immersion resulted in a red-shift of the absorptance peak (Figure 7b). In contrast, in the case of poor solvent immersion, the thickness of the PGMA grating decreased with increasing the immersion time beyond 30 s, resulting in a blue-shift of absorptance peak because of the coiling regime of the PGMA brushes. These observations reveal that the absorptance peak of the grating could be further tuned with stretching and coiling of the regimes through solvent immersions.

Figure 8a shows the solvent dependence of PQ1.5 as peak intensity. Moreover, the surface texture considerably decreased on the grating space due to the collapsing effect of polymer brushes. The intensity of PQ1.5 in the air after dioxane (good solvent) treatment was significantly enhanced, indicating the stretched regime in the line array layer. In contrast, the stretched regime was suppressed by water (poor solvent) treatment, leading to the coiled regime in the line array layer, resulting in the lowest intensity. The re-immersion into the dioxane de-stretches the polymer chains, resulting in an increase of intensity. The intensity is reversibly switchable after treatment with water and dioxane, exhibiting conformational changes in the tethered PGMA. These effects are reversible for the six cycles (Figure 8b). Overall, the results show that the single-layered geometry PGMA-CQD gratings can be simply tuned to attain an intensity range of PL properties. These observations suggest that the fabricated nanosensors possess great potential for sensing the surrounding nature media.

## 4. Conclusions

A “grafting from” system is employed with an ATRP approach to produce PGMA brushes on Si wafers. This process also yields patterned polymeric thin films which act as an absorber in the nanoscale level. 1DPPG, with various scales in geometry, exhibits the spectral absorptance ranged from the wavelength of 371 nm to 459 nm. We immobilized CQDs that possessed a characteristic absorption peak of 395 nm on the 1DPPGs. The results suggest that the fluorescence of QDs could be enhanced by the structure of the grating absorber when the QDs and gratings having the similar absorptance position. The fluorescence of QDs on the grating could be further tuned with swelling-shrinkage of the PGMA brushes using solvent immersions. The proposed approach seems to be versatile in nature, and can be successfully employed to modify various grating surfaces through the immobilization of the different types of QDs. This study also offers the chance to determine the exact methodology in applications of environmental detection when surface modification with polymer brushes is required.

## Figures and Tables

**Figure 1 polymers-11-00558-f001:**
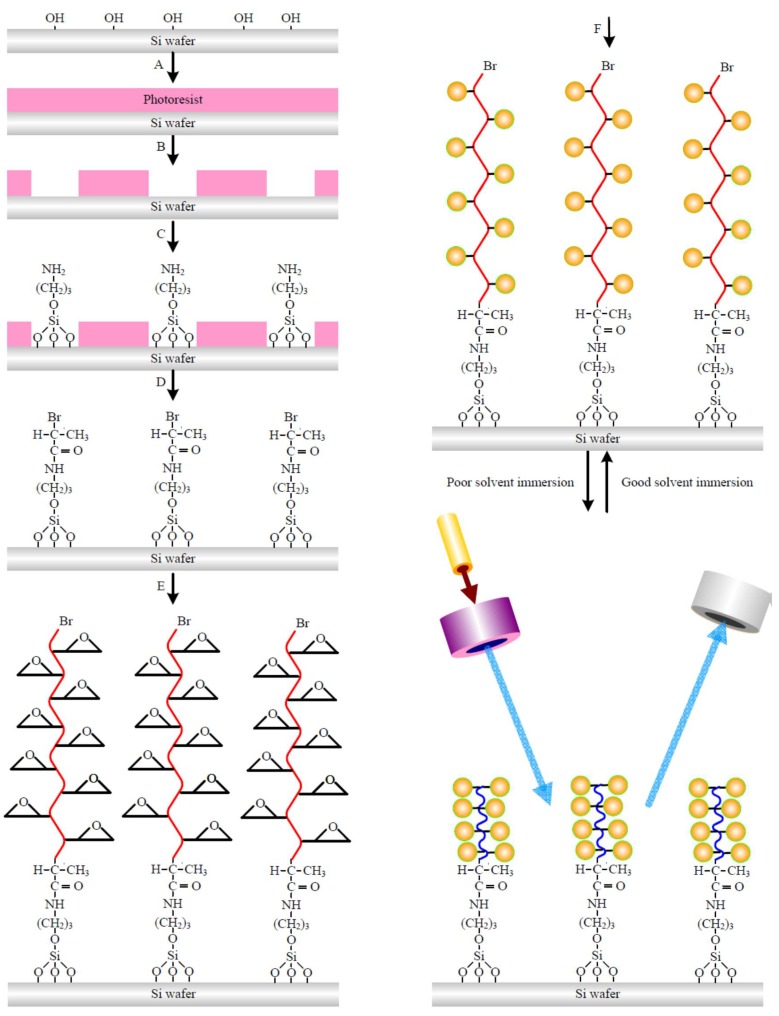
Schematic illustrations of the immobilization of CQDs on the 1DPPG. Si wafers were employed with HMDS in a thermal evaporator. Photoresist was spin coated onto the Si surface (**A**). I-line lithography was used to pattern the photoresist into trench arrays on the surface (**B**). AS selectively assembled on the regions of trench bottoms of Si surfaces (**C**). BB selectively reacted with AS-treated Si surface to form the initiator (**D**). The sample grafting through surface-initiated (ATRP) of GMA from the functionalized areas of the line patterns as 1DPPG (**E**). Immobilization of CQDs on the 1DPPG by covalent bonding exhibits solvent-responsive behavior on optical property (**F**).

**Figure 2 polymers-11-00558-f002:**
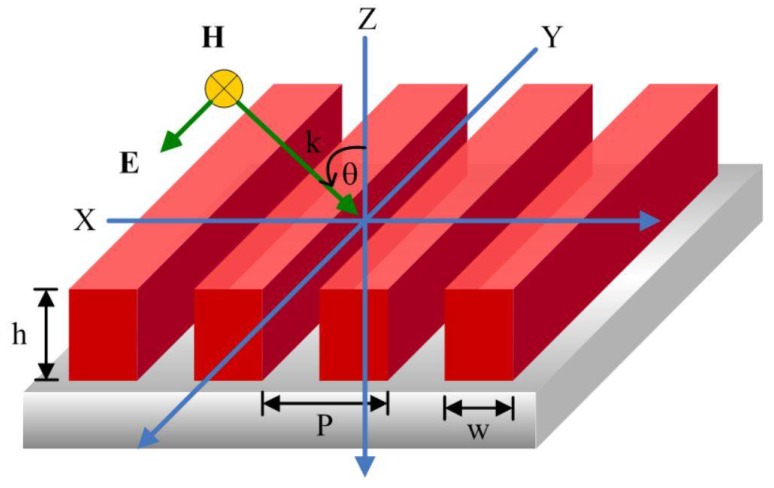
One-dimensional grating having grating period (P), width (w) and thickness (h).

**Figure 3 polymers-11-00558-f003:**
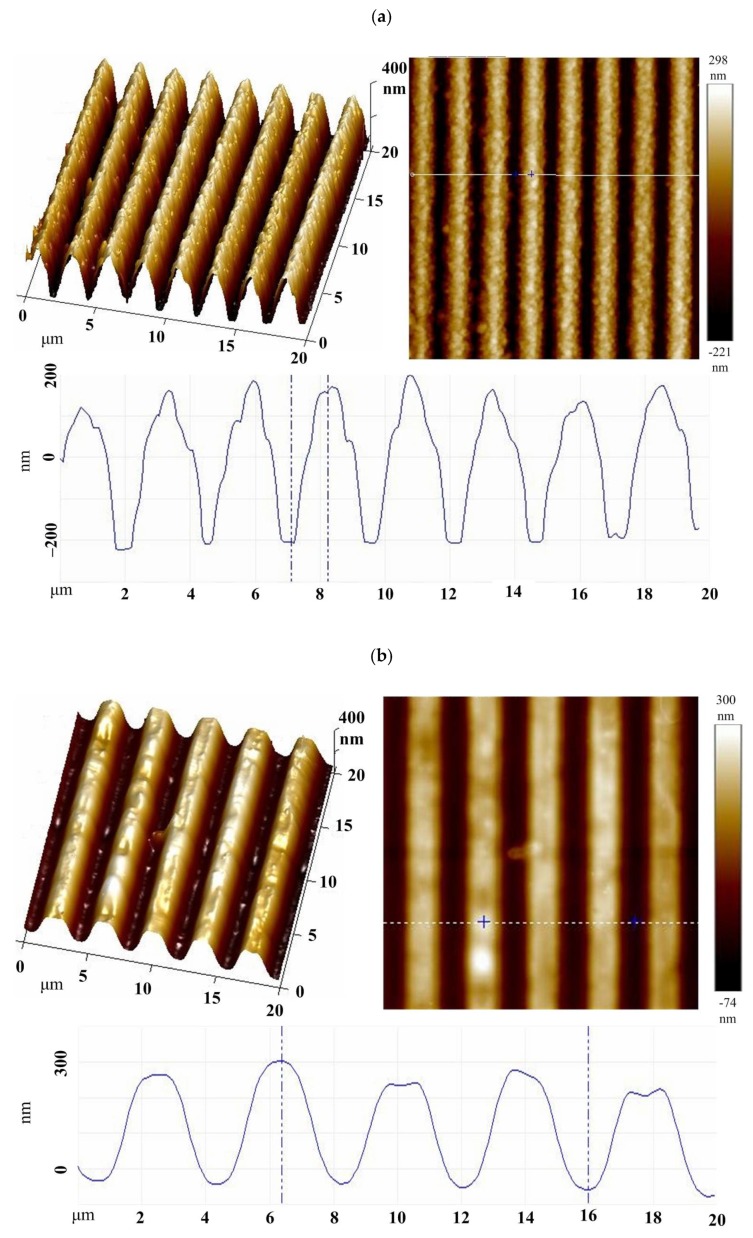
2D and 3D AFM height images (20 μm × 20 μm) of (**a**) PG1, (**b**) PG1.5, (**c**) PG2 and (**d**) PG3, respectively.

**Figure 4 polymers-11-00558-f004:**
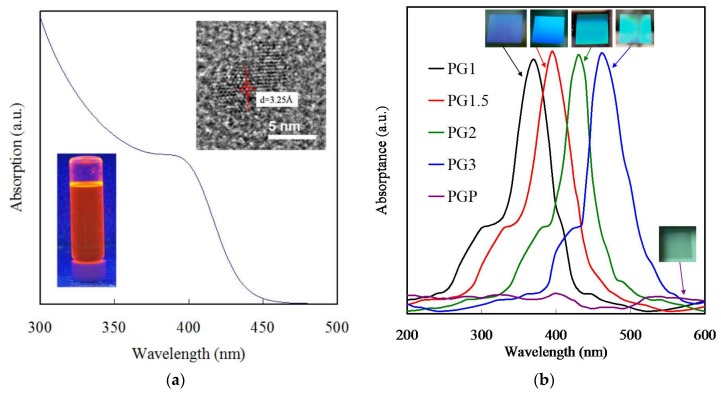
UV–vis reflectance absorptance spectra of (**a**) CQDs, inserts represent the TEM images and photography of CQDs, and (**b**) PG1, PG1.5, PG2, PG3, and planer sample.

**Figure 5 polymers-11-00558-f005:**
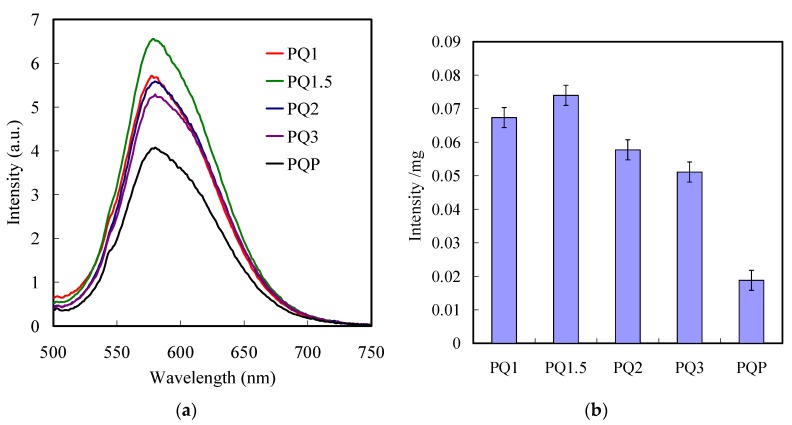
PL emission spectra of (**a**) PQ1, (**b**) PQ1.5, (**c**) PQ2, (**d**) PQ3, and planer sample.

**Figure 6 polymers-11-00558-f006:**
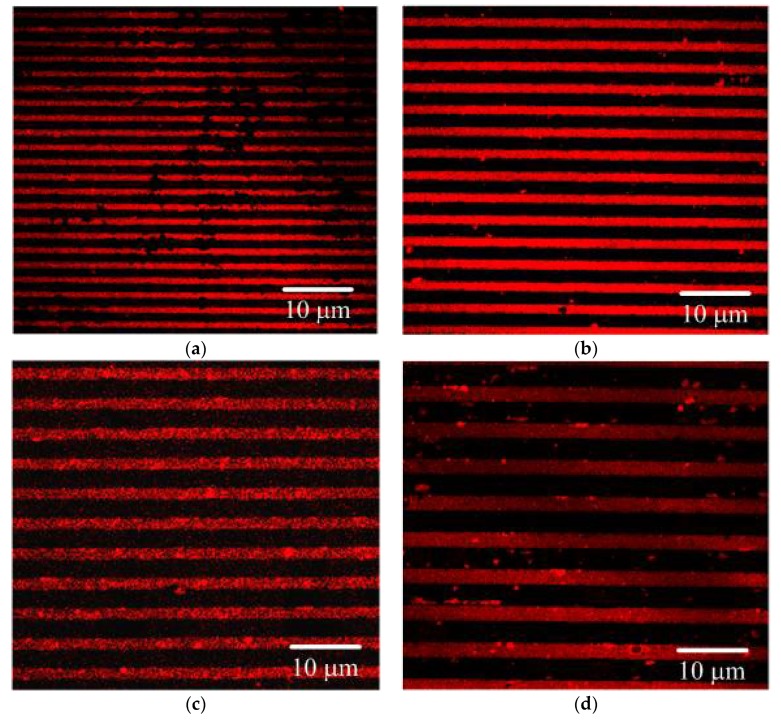
CLSM images of (**a**) PQ1, (**b**) PQ1.5, (**c**) PQ2, (**d**) PQ3, and (**e**) planer sample.

**Figure 7 polymers-11-00558-f007:**
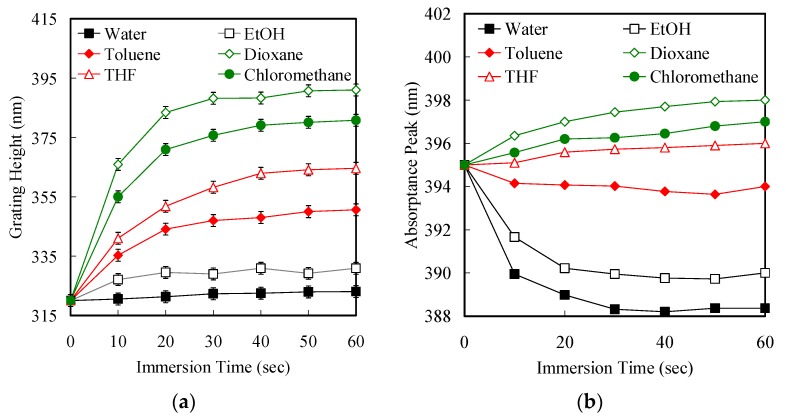
(**a**) Grating height and (**b**) absorptance peak of PQ1.5 plotted with respect to immersion times in various solvents.

**Figure 8 polymers-11-00558-f008:**
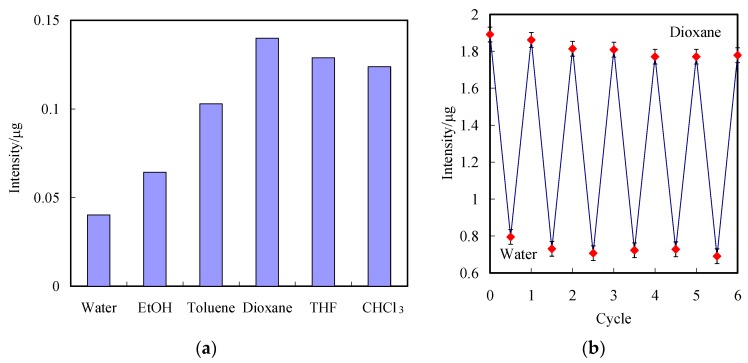
(**a**) Ratios of PL intensity at 395 nm of PQ1.5 after immersion in water, EtOH, toluene, dioxane, THF and CHCl_3_, respectively, to those in air. (**b**) PL intensity of PQ1.5 after six cycles of immersing in water and dioxane.

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
