# Peer review of "Coordination between Surface Lattice Resonances of Poly(glycidyl Methacrylate) Line Array and Surface Plasmon Resonances of CdS Quantum on Silicon Surface"

_polymers, 2019, doi:10.3390/polym11030558_

Round 1

Reviewer 1 Report

This manuscript described the synthesis of the one-dimensional gratings composed of poly(glycidyl methacrylate) (PGMA) brushes and CdS quantum dots (CQDs) and their optical properties, i.e. both surface lattice resonances of PGMA grating and surface plasmon resonances of CQD. The experiment is well designed and the results which are argued by author are interesting. However, there are several problems in this manuscript. In my opinion, this might be interesting after more improvement. Authors should clarify the following issues to strengthen this manuscript.

1. The 1-line lithography was utilized for making the trench arrays with the resolution of 1 μm, 1.5 μm, 2 μm and 3 μm. Is there any reason for choosing the range of these resolutions? What if the resolution is lower than 1 μm or higher than 3 μm?

2. The grating periods of PQ1.5, PQ2 and PQ3 are designed to have a similar thickness (320 nm). According to the Figure 7, it seems that the absorption peak depends on the height of gratings. What if you use the gratings having a thickness higher than 320 nm for the same solvent immersion test? Have you tried to adjust the initial height of gratings to find the optimum point so that you can achieve the optimized optical properties?

3. In Figure 3, the 2D and 3D AFM height images are provided. For the better understanding, the authors should provide color scale bar for each images. In addition, the height graphs should be revised to set the zero (0 nm) as an origin, for the better comparison. Moreover, the authors should provide the roughness of gratings for each types (PG1, PG1.5, PG2) and should explain if there is any effect of the roughness on the optical properties.

4. The PQ1.5 exhibits the highest fluorescence among the PGMA-CQD gratings, since the absorption peak of PQ1.5 is perfectly matched with that of CQDs, facilitating the recombination rate to the carrier. For the better understanding, the authors should complement the reason why the PQ1.5 shows a well-matched absorption peak with CQDs compared to other types, i.e. PQ1, PQ2 and PQ3. More scientific and detailed elucidations are required in the revised manuscript.

5. In good solvents, the thickness of PGMA gratings increases, resulting in the red-shift. In contrast, the thickness of PGMA gratings decreases in poor solvents, leading to the blue-shift. The authors need to clarify which type of solvents are good or poor for the PGMA gratings, so that the readers can clearly understand the principles. More scientific explanations should be strengthened.

6. Lastly, as a minor point,

   1) There is a typo in the title. The author missed a parenthesis in the title.

  Poly(glycidyl methacrylate” Poly(glycidyl methacrylate)”

2) There are some typing errors in the manuscript.

 For instance, there are some cases where hyphen is misused, as follows.

  “-NH3”, “-OH”, “-COOH”, “-SH” → “−NH3”, “−OH”, “−COOH”, “−SH”

 (Page 3, Line 54)

“-40 °C” → “‒40 °C” (Page 6, Line 151)

“Figure 6a-d” → “Figure 6a‒d” (Page 19, Line 375)

  In addition, spacing of the words in some sentences is wrong, as follows.

“1μm, 1.5μm, 2μm” → “1 μm, 1.5 μm, 2 μm” (Page 4, Line 81)

  “80°C” → “80 °C” (Page 5, Line 131)

  Other typing errors: “Fig 2” → “Figure 2” (Page 9, Line 196)

  3) Some words should be written in italics, as follows.

“i.e.” → “i.e.(Page 9, Line 182)

“i.e.” → “i.e.(Page 9, Line 191)

“ca.” → “ca.(Page 16, Line 307)

“ca.” → “ca.(Page 16, Line 317)

Accordingly, I recommend this manuscript for publication in Polymers after major revision.

Author Response

Dear Editor/Reviewer:

Thank you for reviewing our manuscript and providing valuable comments. Your precious information is greatly appreciated. I have listed below our replies to your comments.

Reviewer #1: Comments and Suggestions for Authors

This manuscript described the synthesis of the one-dimensional gratings composed of poly(glycidyl methacrylate) (PGMA) brushes and CdS quantum dots (CQDs) and their optical properties, i.e. both surface lattice resonances of PGMA grating and surface plasmon resonances of CQD. The experiment is well designed and the results which are argued by author are interesting. However, there are several problems in this manuscript. In my opinion, this might be interesting after more improvement. Authors should clarify the following issues to strengthen this manuscript.

1. The 1-line lithography was utilized for making the trench arrays with the resolution of 1 μm, 1.5 μm, 2 μm and 3 μm. Is there any reason for choosing the range of these resolutions? What if the resolution is lower than 1 μm or higher than 3 μm?

Our reply: Scale lower than 1, 1.5, 2 and 3 um should be better according to effective medium theory. However, lithography system to fabricate these arrays at lower scale are much more expensive. We did not have such high level lithography system. Therefore, line arrays with 1, 1.5, 2 and 3 μm scale were fabricated in this work. We are planning to establish the high level lithography system recently.

2. The grating periods of PQ1.5, PQ2 and PQ3 are designed to have a similar thickness (320 nm). According to the Figure 7, it seems that the absorption peak depends on the height of gratings. What if you use the gratings having a thickness higher than 320 nm for the same solvent immersion test? Have you tried to adjust the initial height of gratings to find the optimum point so that you can achieve the optimized optical properties?

Our reply: Fabrication of the line arrays was to follow a protocol of semiconductor processes. Tuning the thickness of photoresist needs to tune spin-coating rate, baking temperature, development time etc. to obtain the good uniformity of pattern. Therefore, all parameters, such as thickness, temperature, spin-coating rate etc., of the protocol are always invariable to maintain the reliability. Therefore, the thickness of grating could not be changed casually to optimize the optical properties. In addition, because photoresist could be dissolved by solvents, which may not survive in the solvent immersion test. Accordingly, we fabricated the polymer brush grating to investigate the optical properties in the solvent immersion test.

3. In Figure 3, the 2D and 3D AFM height images are provided. For the better understanding, the authors should provide color scale bar for each images. In addition, the height graphs should be revised to set the zero (0 nm) as an origin, for the better comparison. Moreover, the authors should provide the roughness of gratings for each types (PG1, PG1.5, PG2) and should explain if there is any effect of the roughness on the optical properties.

Our reply: We have provided the color scale bar for each image. The origin point was adjusted automatically to obtain the optimal images during the AFM measurement. We had set the origin to zero, however, the quality of AFM images was poor. Generally, roughness is used to define a surface properties without regular pattern, while only surface with regular pattern can exhibit optical properties. Roughness is calculated with the parameters including distribution, height, and direction of pattern, which may not clearly establish the correlation with optical properties. In our case, direction of grating is invariable. Height and distribution are used to establish the correlation with optical properties, which should be better to understand the effects on the optical properties precisely.

4. The PQ1.5 exhibits the highest fluorescence among the PGMA-CQD gratings, since the absorption peak  of PQ1.5 is perfectly matched with that of CQDs, facilitating the recombination rate to the carrier. For the better understanding, the authors should complement the reason why the PQ1.5 shows a well-matched absorption peak with CQDs compared to other types, i.e. PQ1, PQ2 and PQ3. More scientific and detailed elucidations are required in the revised manuscript.

Our reply: Plasmonic gratings have been combined with organic dyes and semiconductors or dipolar emitters for experimental and theoretical study of the spontaneous emission in coupled hybrid systems [29]. The emission of dye molecules can be strongly enhanced and directed in such systems when they radiatively decay through the collective grating modes [30]. Furthermore, stimulated emission in such hybrid systems has also been observed [31]. In addition to the emission enhancement, the coupling of lattice resonances to molecules has also been studied theoretically and experimentally [32,33]. We fabricated these gratings with scale range from 1 to 3 um according the pervious reports to match the absorption peak of quantum dots. The scientific and detailed elucidations have been reported in these articles. The paragraph has been included in the manuscript.

5. In good solvents, the thickness of PGMA gratings increases, resulting in the red-shift. In contrast, the thickness of PGMA gratings decreases in poor solvents, leading to the blue-shift. The authors need to clarify which type of solvents are good or poor for the PGMA gratings, so that the readers can clearly understand the principles. More scientific explanations should be strengthened.

Our reply: Polymers can be well-dissolved in their good solvents that are able to stretch the polymer chains completely, but on the other hand, polymers can not be dissolved in the poor solvent resulting in the polymer collapsing chain [39] In this case, CHCl3, THF, dioxane and toluene are good solvents, while water and ethanol are poor solvents for PGMA. We have addressed the solvent type in the manuscript.

6. Lastly, as a minor point,

1) There is a typo in the title. The author missed a parenthesis in the title.

  Poly(glycidyl methacrylate Poly(glycidyl methacrylate)

Our reply: The typo has been corrected.

2) There are some typing errors in the manuscript.

 For instance, there are some cases where hyphen is misused, as follows.

  “-NH3”, “-OH”, “-COOH”, “-SH” “−NH3”, “−OH”, “−COOH”, “−SH”

 (Page 3, Line 54)

 -40 °C “‒40 °C” (Page 6, Line 151)

 Figure 6a-d” “Figure 6a‒d” (Page 19, Line 375)

Our reply: All typos have been corrected.

  In addition, spacing of the words in some sentences is wrong, as follows.

1μm, 1.5μm, 2μm 1 μm, 1.5 μm, 2 μm (Page 4, Line 81)

  80°C 80 °C (Page 5, Line 131)

  Other typing errors: Fig 2” Figure 2” (Page 9, Line 196)

Our reply: All typos have been corrected.

  3) Some words should be written in italics, as follows.

i.e. i.e. (Page 9, Line 182)

i.e. i.e. (Page 9, Line 191)

ca. ca. (Page 16, Line 307)

ca. ca. (Page 16, Line 317)

Our reply: All items have been written in italics.

Thank you for reviewing our manuscript and providing valuable comments. Your precious information is greatly appreciated.

I am looking forward to hearing from you soon.

Thanks very much.

Sincerely,

Best regards,

Prof. Jem-Kun Chen

Department of Polymer Engineering, National Taiwan University of Science and Technology, 43, Sec 4, Keelung Rd, Taipei, 106, Taiwan, R.O.C.

Email : jkchen@mail.ntust.edu.tw

TEL: 886-2-27376523

FAX: 886-2-27376544

Reviewer 2 Report

·         There is a typographical error in the title itself

 “poly(glycidyl methacrylate)” instead of “poly(glycidyl methacrylate”

·         In the abstract section, the author needs to improve with more specific findings and academic advantages.

·         There is lack of continuity in the introduction part and the authors have to rewrite accordingly. Also need to strength the present system, the authors have to cite the recent bibliographies in the introduction section. Several abbreviations are not explained properly where it has used in first time. It should be corrected.

·         The absorption peak (hump) around 280 nm belongs to which transition in Figure 4(b)? The author have to provide the inset picture of planer sample individually in Figure 4(b)

·         The author needs to correct the typo-error in Figure 8(b).” Dioxane” instead of “Dioxan”

·         There are numerous typographical and language errors, so the proper English corrections are required to carried out.

Author Response

Dear Editor/Reviewer:

Thank you for reviewing our manuscript and providing valuable comments. Your precious information is greatly appreciated. I have listed below our replies to your comments.

Reviewer #2: Comments and Suggestions for Authors

· There is a typographical error in the title itself

 “poly(glycidyl methacrylate)” instead of “poly(glycidyl methacrylate”

Our reply: The typo has been corrected.

In the abstract section, the author needs to improve with more specific findings and academic advantages.

Our reply: We have improved the abstract with more specific findings and academic advantage.

There is lack of continuity in the introduction part and the authors have to rewrite accordingly. Also need to strength the present system, the authors have to cite the recent bibliographies in the introduction section. Several abbreviations are not explained properly where it has used in first time. It should be corrected.

Our reply: We have rewritten the introduction to improve the continuity.

The absorption peak (hump) around 280 nm belongs to which transition in Figure 4(b)?

Our reply: A grating usually absorbs several wavelengths. In this case, the gratings also absorb 280 nm wavelength partially, but the degree of absorptance at 280 nm was lower than that at 370 nm for PG1. Therefore, we select the absorptance at 370 nm as the marker to identify the shift of absorptance with scale for the stripe pattern. The grating only absorbs characteristic wavelength without transition.

The author have to provide the inset picture of planer sample individually in Figure 4(b)

Our reply: We have provide the insert picture of planar sample individually in Figure 4b. The inset to Figure 4b presents photographic images of the PG1, PG1.5, PG2, PG3 and PGP, recorded using a common camera. As-prepared PG1 exhibited a blue color, which could be regarded as a resonant absorber. Upon increasing the scale, the films gradually turned to light green. The PFP film appeared slightly off-white without any color visualizing, indicated that the flat film did not possess any effect of resonant absorber with lower absorption intensity.

The author needs to correct the typo-error in Figure 8(b).” Dioxane” instead of “Dioxan”

Our reply: The typo has been corrected.

There are numerous typographical and language errors, so the proper English corrections are required to carried out.

Our reply: The manuscript has been edited by a native English speaker.

Thank you for reviewing our manuscript and providing valuable comments. Your precious information is greatly appreciated.

I am looking forward to hearing from you soon.

Thanks very much.

Sincerely,

Best regards,

Prof. Jem-Kun Chen

Department of Polymer Engineering, National Taiwan University of Science and Technology, 43, Sec 4, Keelung Rd, Taipei, 106, Taiwan, R.O.C.

Email : jkchen@mail.ntust.edu.tw

TEL: 886-2-27376523

FAX: 886-2-27376544

Round 2

Reviewer 1 Report

The changes the authors have made are satisfactory. The answers to the comments and requirements were clarified and strengthened in the revised manuscript. Accordingly, I recommend this manuscript for publication in Polymers without further revisions.